GoPros™ as an underwater photogrammetry tool for citizen science

Raoult Vincent vincent.raoult@mq.edu.au
David Peter A.
Dupont Sally F.
Mathewson Ciaran P.
O’Neill Samuel J.
Powell Nicholas N.
Williamson Jane E.
Biological Sciences, Macquarie University , Sydney NSW , Australia
Pawlik Joseph
Electronic publication date: 2016 Apr 25
Publication date: 2016
Volume: 4
Electronic Location ID: e1960
Received 2016 Feb 25; Accepted 2016 Apr 1
Copyright: ©2016 Raoult et al.
Copyright year: 2016
Copyright holder: Raoult et al.
License: This is an open access article distributed under the terms of the Creative Commons Attribution License, which permits unrestricted use, distribution, reproduction and adaptation in any medium and for any purpose provided that it is properly attributed. For attribution, the original author(s), title, publication source (PeerJ) and either DOI or URL of the article must be cited.
License URL: https://creativecommons.org/licenses/by/4.0/

Keywords: Photogrammetry, Structure-from-motion, Transects, Coral health, Coral cover, Diversity, Coral reef

Funding: Macquarie University University of Queensland Funding was provided by Macquarie University, and facilities were provided by the University of Queensland. The funders had no role in study design, data collection and analysis, decision to publish, or preparation of the manuscript.

==============================
Citizen science can increase the scope of research in the marine environment; however, it suffers from necessitating specialized training and simplified methodologies that reduce research output. This paper presents a simplified, novel survey methodology for citizen scientists, which combines GoPro imagery and structure from motion to construct an ortho-corrected 3D model of habitats for analysis. Results using a coral reef habitat were compared to surveys conducted with traditional snorkelling methods for benthic cover, holothurian counts, and coral health. Results were comparable between the two methods, and structure from motion allows the results to be analysed off-site for any chosen visual analysis. The GoPro method outlined in this study is thus an effective tool for citizen science in the marine environment, especially for comparing changes in coral cover or volume over time.

Introduction

The assessment of habitats to understand the demography of animal and plant distributions in relation to perturbations is a central theme in ecology (Guisan & Zimmermann, 2000). Such analyses rely heavily on the accurate spatial resolution of habitats and their inhabitants to generate ecologically relevant predictive hypotheses on community drivers. While accurate spatial methodologies have been generated for terrestrial habitats (Belay et al., 2015), assessments of aquatic habitats are often confounded due to the difficulties of examining large areas under water (Leonardsson, Blomqvist & Rosenberg, 2016), causing researchers to focus on predictive modelling rather than surveying (Rengstorf et al., 2013).

Aquatic habitats of particular concern are coral reefs, which are under threat from multiple anthropogenic sources (Smith et al., 2013). Such threats include warming waters (Doney et al., 2012), ocean acidification (Hooidonk et al., 2014), eutrophication/pollution (Jessen et al., 2013; Koop et al., 2001), and overfishing (Loh et al., 2015). As a result, accurate and precise large-scale assessments of global health, growth and diversity of coral reef habitats are necessary to further assess and monitor such impacts. Assessments require repeated and widespread reef habitat surveys, which have traditionally been done using trained snorkelers (De’ath et al., 2012). As global assessments of such habitats are too time-intensive and costly to be done by any single research group, large-scale data are preferentially obtained through the expansion of citizen science whenever possible (Dickinson et al., 2012; Dickinson, Zuckerberg & Bonter, 2010; Foster-Smith & Evans, 2003).

Citizen science generally refers to scientific research done in part or in whole by collaborators that lack credentials or formal training in the area of expertise (Cohn, 2008). While the practice of using collaborators for science is not new, the process has substantially expanded to include less specialised collaborators with the wider adoption of the internet, which has simplified information exchange over broader scales (Dickinson, Zuckerberg & Bonter, 2010). One of the primary issues with citizen science is the need for ongoing training of the collaborators, without which the quality of the data generally suffer (Dickinson et al., 2012). In these situations, methodologies that are simplified are more reliable and thus more likely to produce rigorous datasets as they require less training. Citizen science initiatives with basic methodologies, such as CoralWatch (Marshall, Kleine & Dean, 2012), are successful because individuals are only asked to match coral colours to a supplied colour chart and do not need to make subjective decisions based on prior knowledge of the system. Despite their success, however, such simplified methodologies often have inherently fewer variables than studies exclusively performed by scientists. A citizen science approach that still captures large amounts of accurate data, therefore, would be beneficial.

Over the last few decades, researchers have been streamlining reef surveying techniques by using novel technologies such as video or photographic analysis of data (Lam et al., 2006; Parker Jr, Chester & Nelson, 1994; Rogers & Miller, 2001). These techniques are generally expensive, requiring specialised video equipment, and rely on experienced divers and snorkelers. More recently, low-cost action cameras that have depth tolerances equal to or below that of recreational diving have become highly popular, the most prominent of which is the GoPro™, which sold over five million units in 2014. Due to their relative ease of use and high resolution/low-cost ratio (Gintert et al., 2012), a number of novel methodologies that use action cameras for marine research have emerged (Assis et al., 2013; Harasti et al., 2014; Letessier et al., 2015). Analysing video footage for scientific purposes, however, remains time-consuming and generally requires expert examination of each frame of video.

Structure from motion (SFM) is a novel image processing technique that allows accurate three-dimensional models and textures to be calculated from a video or series of two-dimensional photos (Westoby et al., 2012). SFM first requires the photos to be aligned, a process that is often aided by GPS tagged images, and the aligned photographs can then be used to create a 3D point map: the points that can be tracked on multiple aligned images are used to create coordinates in a 3D environment. Once a dense point map has been created, a mesh can be created between the points and textured like any 3D model whether this is a small object or a large terrain map (Mancini et al., 2013). Accurate volume/length measurements can then be taken from the 3D reconstruction using photogrammetry: the use of corrected photography as a mapping tool to measure objects within the survey area (Figueira et al., 2015). Traditional applications of photogrammetry were developed for robotics to facilitate manipulation of objects (Yuan, 1989). Digital cameras with the requisite resolution have historically been expensive and not widely available. As such, the methodology has largely been constrained to geological surveys with large budgets (Drury & Drury, 2001). Recently, however, the proliferation of cheaper high-megapixel cameras, affordable unmanned aerial vehicles, and more powerful personal computers have made SFM/Photogrammetry more widely accessible, allowing its use in diverse fields such as palaeontology (Falkingham, Bates & Farlow, 2014), forestry (Bohlin, Wallerman & Fransson, 2012), or fisheries biology (Rohner et al., 2011). Recreational use of these techniques for exploration has been growing, with community projects that include historical shipwreck mapping or modelling of tourist sites. Moreover, there is an increasingly large pool of enthusiasts that have the capabilities to conduct reliable SFM on a variety of subjects, and that have the desire to do so.

This study assesses the use of citizen scientists with GoPros in conjunction with off-site SFM to conduct a typical habitat sampling method for marine ecologists: strip transects (McCormick & Choat, 1987). A coral reef flat was used as the test habitat because it is relatively structurally complex in comparison with many other marine habitats, which has distinct clusters of organisms interspersed in a relatively homogeneous environment, and is commonly sampled using strip transects. We compared our novel sampling regime in this habitat with the traditional snorkel survey technique. A number of commonly used factors were analysed to compare and contrast the two methods: (1) variation in species count of sessile organisms (often used as a measure of biodiversity); (2) variation in benthic cover of corals and macroalgae; (3) mean abundance of holothurians (used here as an indication of resolution); (4) applicability of use within the CoralWatch reef health surveying program (Marshall, Kleine & Dean, 2012); (5) time taken to perform each analysis; and (6) the rate of learning each methodology for new citizen scientists.

Materials and Methods

Study site

This research was conducted under Great Barrier Reef marine Park Permit number QC14/004. Surveying was done at the southern reef flats on Heron Island (23.4420′S, 151.9140′E), a coral cay at the southern end of the Great Barrier Reef, Australia. This reef flat is characterised by patchy sand, coral and algal communities, and is isolated from the ocean during low tide (Vacher & Quinn, 1997). Data collection occurred from 29/03/15 to 3/04/15 during mid to high tides when water was sufficiently deep to snorkel (1–2 m depth). Southern Heron reef flat has been studied extensively, and at that time of the year is dominated by the coral genus Acropora (Santos et al., 2011), and the macroalgal species Padina gymnospora, Caulerpa racemose clavifera and Sargassum polycystum (Cribb, 1966; Scopélitis et al., 2011).

Snorkel surveys

Twelve transects were haphazardly placed around the reef flat to compare surveys that used GoPro cameras and traditional snorkel methods. Transects were 50 m long and 2 m wide, or approximately 100 m2, and were separated by at least 50 m. Each transect was surveyed twice (one benthic survey, one holothurian survey: these two surveys were separated to simplify the process for snorkelers, though experienced surveyors could easily conduct both at once in other circumstances) by five independent snorkelers who had taken a two-day course on snorkel safety and coral/algae identification, similar to training courses for prospective citizen scientists (Foster-Smith & Evans, 2003). To reduce bias that may be caused by remembering the composition of habitats, the traditional snorkel surveys (Hill & Wilkinson, 2004) and GoPro recording swims were conducted in alternating order.

Gopro photogrammetry/structure from motion surveying

GoPro transects were surveyed only once. To ensure comprehensive coverage of the area, one snorkeler covered the transect area twice (the length of the transect and back) using a GoPro Hero 3 Black™ that was set to continually capture images at maximum resolution at 2 Hz (0.5 images per second setting). The benefit of GoPros for this application is their wide view angle, which allows users to cover a larger area per image, however, any action camera (potentially more affordable) with a wide viewing angle and high pixel density would suit this application. Snorkelling was done at a slow pace with the arm holding the GoPro outstretched close to the surface and the GoPro aimed straight down at the substrate. Slow swim speeds are necessary as GoPros and most other digital cameras are CMOS-based and have a rolling shutter, which can cause the deformation of images if the objective is moving too quickly (Chun, Jung & Kyung, 2008). The aim of the snorkeler was to get over 60% overlap from pictures to ensure they could be aligned, and preliminary testing indicated this method decreased alignment errors over single passes or higher image intervals. GoPro transects were done towards high tides when possible to increase the area of the GoPro’s coverage.

Image analysis

An orthocorrect (with corrected distances between points) 3D textured model that could be used to conduct virtual surveys through structure from motion and photogrammetry was created (Falkingham, Bates & Farlow, 2014; Westoby et al., 2012). Images for each transect were compiled using Agisoft Photoscan (Agisoft LLC), a 3D photogrammetry program that can build point meshes and orthomosaics from digital photos. While open access SFM programs exist (i.e., VisualSFM combined with Meshlab), no open access software combines all the functions of the professional edition of Agisoft Photoscan (image correction, image alignment, mesh creation, texture rending, volumetric measurements, and network processing). The standard edition of Agisoft Photoscan does not include the network processing or measurement functions, both aspects that would be relevant to scientific applications (network processing would allow rapid processing of transects, for instance). For users who are not familiar with SFM, we highly recommend this program until comparable open access software is available. Those who are more comfortable with troubleshooting, however, can obtain similar results using current open access software. Photographs were automatically corrected for lens distortion by selecting the ‘GoPro’ image correction option. Photographs were aligned using assumed pairing (the software assumes successive photographs were taken close to each other). From this alignment, Agisoft Photoscan then renders a dense point map, which places corresponding points in overlapping photos into three-dimensional space. High complexity depth maps were reconstructed from the point map, where the orientation of the mesh was established (approximately 1,000,000 polygons). Photographic textures were then laid over the mesh, and a high-quality orthomosaic was computed. The resulting TIFF files were approximately 60,000 × 10,000 pixels (a resolution of ∼1.2 mm per pixel). The inputs required to produce such an image only include selecting the images and the appropriate analyses (under 5 min to conduct), however, processing time can vary greatly depending on the photograph content and the processing power of the computer. This last step (creation of a photomosaic) is not necessary for image analysis but was added so that results could be more accessible to readers of this manuscript.

Benthic survey

The benthic survey estimated bommie cover, biodiversity of corals (to genus) and macroalgae (to species). A preliminary survey allowed the reliable identification of predominant macroalgal species, and corals were identified to genus using the CoralFinder™ (a waterproof coral identification handbook) (Kelley, 2009) and a priori training. Training was considered complete when snorkelers consistently identified corals to the same genus. Using the CoralFinder’s ruler as a reference during snorkelling and the transect line during image analysis, the size of bommies were binned according to approximate minimum radius categories (Marsh, Bradbury & Reichelt, 1984): (a) from 20 to 50 cm radius, and (b) greater than 50 cm radius.

Variance comparisons are often used when comparing different surveying methodologies (Harvey et al., 2004; Watson et al., 2005; Willis, Millar & Babcock, 2000). While it is likely that different methods are better at surveying certain aspects of the environment and may result in different means, similar rates of variation indicate that different methods are as reliable and are as likely to produce consistent results. A paired t-test was therefore used to compare mean variance of traditional benthic survey snorkelling and GoPro photogrammetry, with the aim of determining whether GoPro photogrammetry could be used as an alternative technique to snorkel surveys.

Holothurian abundance

Holothurians are a prominent element of shallow tropical sediments between bommies on coral reefs. Counts of holothurians occurring in the transects using both traditional and GoPro methods were done. No attempt was made to separate individuals into species, and abundance was purely a count of the number of individuals observed in a transect. Individuals were included in the transect count if any part of their body was visible, thus, counts here may be a slight over-estimation of real abundance. As we were using the counts for comparative methods only, this did not matter. A nested ANOVA was used to compare abundance counts between GoPro photogrammetry and traditional snorkel methods(fixed factor) with transect number as a random factor, and to consider whether there were significant variations between observers (nested in snorkel method).

Coral health

Coral health assessments are a necessary means of monitoring the progress of anthropogenic impacts on coral reefs (Hodgson, 1999). CoralWatch is a global citizen science project that facilitates effective reef management by assessing coral health by using a standardised health chart (Marshall, Kleine & Dean, 2012). Coral health is inferred by associating the colour of corals with the presence or absence of symbiotic zooxanthellae, the lack of which is characteristic of stressed or dying coral (Rosenberg et al., 2007). By using the simplified chart, members of the community rather than scientists can assess reefs under their care thus empowering and fast tracking early signs of reef decline.

To assess whether our GoPro photogrammetry methods could be used to assess coral health via CoralWatch, twenty corals were chosen at random along each of the transects and their colours assessed using the standardised chart. Lightest and darkest colours, as well as coral type (branching, boulder etc.) were recorded for each of the chosen corals as per Siebeck, Logan & Marshall (2008). Corals were assessed using charts on traditional snorkel methods (as per CoralWatch practice) then compared with chart use on the images produced via the GoPro transects. Two-tailed t-tests were used to compare the mean lightest and darkest colour between GoPro photogrammetry transects and snorkel transects.

Figure 1 Example of transect produced using GoPro structure from motion.

Here is an example of one of the transects in this study constructed using GoPro structure from motion. Note that the resolution of the source file has been greatly lowered to make it more accessible: the original file had a resolution of 60,000 × 10,000 pixels and a size of ∼50 mb.

Sample time comparison

One of our expectations of our GoPro photogrammetry surveying was that the method would reduce time taken in the field as assessments of orthomosaics can be done later in a more comfortable lab environment. We predicted that the time to complete GoPro photogrammetry surveys would be lower than that of the traditional snorkelling method. We also predicted that the time to complete traditional snorkelling transects would improve at a greater rate and be longer initially than structure from motion surveys with newly trained citizen scientists because these individuals would need to make taxonomic decisions on site. The time taken to complete transects was recorded using stopwatches to the second. The time taken to complete benthic surveys, holothurian counts and CoralWatch surveys were timed separately for both snorkel surveys and for GoPro photogrammetry for each observer. Observer transect times for each transect were then compared using a generalised linear model to estimate the rate of improvement for both methods. A nested ANOVA was used to compare the time to complete each survey between traditional transects and GoPro photogrammetry once improvement effects were corrected (the observers were the nested factor, the transect number the random factor, and the method was a fixed factor). Time taken for image analysis was not included because it is performed exclusively by the computer software and requires no hands-on input. Commencing the analysis itself takes under five minutes per transect, and image processing is highly dependent on the number of images taken and the required point density (higher is better for volumetric calculations). In this study, with ∼500 images per transect, processing time was roughly two hours per transect (again, this would be variable depending on the complexity of the habitat).

Results

Orthomosaics could be produced from the GoPro surveys. Figure 1 shows an example of an orthomosaic produced from one of the GoPro structure from motion surveys (a full, high-resolution version is available at https://figshare.com/s/5a1644840d1311be5137). A supplemental video is also available for a moving ‘flight’ over the same transect, but using a high polygon 3D model produced from structure from motion (https://figshare.com/s/d429569435f3ba970e3b). ‘Bommies’ (clusters of corals and macroalgae), interspersed amongst homogeneous sand, are clearly visible in this moving ‘flight.’ Holothurians are also visible on the sand and near the bommies. This method produced approximately 550 12 megapixel images per transect.

Figure 2 Mean benthic diversity per transect.

Species diversity box plots comparing traditional snorkelling and GoPro photogrammetry.

Benthic diversity

Mean species diversity of corals and algae was significantly different between the two methods, with snorkel transects showing richer diversity than SFM transects (t = 7.104, df = 111, p < 0.001). GoPro photogrammetry and traditional snorkel surveys were similar in their assessments of benthic diversity, as the variance of species diversity was not significantly different between the two (t = 1.605, df = 11, p = 0.068, Fig. 2).

Benthic cover

Mean bommie cover estimates were not significantly different between snorkelling and GoPro photogrammetry (t = 1.36, df = 22, p = 0.18). Furthermore, GoPro photogrammetry and traditional snorkel surveys were not significantly different in their assessments of benthic cover, as the variance was not significantly different between the two (t = 0.88, df = 11, p = 0.199, Fig. 3).

Figure 3 Mean benthic cover per transect.

Benthic cover (coral bommies) box plots comparing results from traditional snorkel surveys from GoPro photogrammetry.

Holothurian abundance

Estimates of holothurian abundance were not significantly different for the snorkel transects or the GoPro photogrammetry transects (f = 4.253, df = 1, p = 0.042, Fig. 4).

Figure 4 Mean holothurian count per transect.

Mean holothurian count per transect box plot, comparing snorkel and GoPro structure from motion.

Coral health

Using the CoralWatch health chart, GoPro photogrammetry transects were significantly darker than snorkel transects for the lighter colours (t = 7.89, df = 59, p < 0.001). There was no significant difference between GoPro photogrammetry and snorkel transects for the darker corals (t = 1.98, df = 118, p = 0.15, Fig. 5).

Figure 5 Mean Coralwatch values.

Mean ± 1. S.E. lightest and darkest CoralWatch colour chart results using either traditional snorkel surveys or GoPro photogrammetry. Asterisks represent means that are significantly different.

Time taken

Structure from motion transects took significantly less time to complete in the field than snorkel transects (f = 17.12, df = 1, p < 0.001). There were no significant differences in mean transect times between observers (f = 1.33, df = 4, p = 0.26), however, snorkel transects appeared to have a larger number of outliers with high times to complete. While mean transect times were significantly lower for SFM transects, the rate of improvement over time was significantly higher for traditional snorkel transects (W = 15.97, df = 1, p < 0.001, Fig. 6). Both methods had an asymptotic improvement trend, and the rate of improvement appeared to be very low or null after the 12th replicate (Fig. 6).

Figure 6 Mean time taken per transect.

Mean ± 1 S.E. time taken per transect across the five observers and between traditional snorkel surveys and GoPro photogrammetry.

Discussion

Structure from motion transects created using GoPro imagery were successfully used to estimate a host of variables that are often measured using traditional snorkelling. The variability of the results was similar using both techniques, suggesting that structure from motion transects are as reliable as traditional snorkelling. This suggests that benthic transects conducted using GoPro and processed through SFM are a viable alternative to traditional snorkel transects for citizen science.

Both GoPro photogrammetry and traditional snorkelling had strengths and weaknesses: while variances were not significantly different, measured benthic (coral and macroalgal) diversity estimates were significantly lower when using GoPro photogrammetry, possibly because of the limited resolution of the images. Doubling the image resolution (0.6 pixels per mm) may solve this issue as it would quadruple image size. This would, however, result in more computer resource use during analysis. Using a camera with a higher-quality lens or sensor (i.e., DSLR) may also solve this issue, however, because this methodology was designed for citizen science, expensive DSLRs and dive housings were not used. Despite the limited ability of structure from motion for the identification of coral polyps (due to limited image resolution), estimates of benthic diversity were as reliable using structure from motion as from traditional snorkelling. Studies interested in pooling results obtained using this method as well as snorkel transects, however, must correct for the lower benthic diversity detected using GoPro photogrammetry. Transforming the data obtained through GoPros by a factor comparing the mean diversity from a snorkel transect in a similar environment (i.e., in this study it would be 1.33) would assist in resolving this issue but needs to be validated in the field for the particular habitat.

Counts of holothurian abundance were not significantly different between traditional snorkelling and GoPro photogrammetry methods. Methodology in our photogrammetry method could be improved, however, and studies that target holothurian abundance and diversity should amend our method to use a slow sweeping recording motion using varied GoPro angles to ensure that the areas below corals are effectively covered. A SFM model, not an orthomosaic, should also be used to count holothurians. Structure from motion algorithms remove ‘moving points’ (i.e., fish or other animals that move at a respectable pace) from images as aberrations, and it is possible the detectability of some holothurians was reduced during image analysis as a result.

Health assessments of coral reefs using the prescribed CoralWatch method and using GoPro photogrammetry were skewed towards the darker colour spectrum in comparison to snorkel surveys. Studies that rely on colour spectrums may need to take precautions and early validation should they wish to use GoPro photogrammetry. Colour differences between GoPro photogrammetry methods and snorkel surveys could be corrected via data transformation (colours were skewed by a mean of 0.43 points) or via image colour correction. Colour correction should also be considered if the colour is a variable that may influence data analysis, especially on darker sampling days or at greater depths. Blue-shift can be corrected either with a red filter on the camera itself (currently available for GoPros) or with image post-processing software such as Adobe Photoshop that includes auto-colour correction functions. Image dehazing, a common technique used to increase the clarity and colour of underwater photography (Chiang & Chen, 2012), could also be used to reduce the issue of colour shift. In this study, the simple addition of including a CoralWatch colour chart on the substratum within our imaged transects would allow an unbiased appraisal of the colour without the need for colour correction.

While structure from motion surveys were significantly faster to complete than traditional snorkelling methods, these analyses did not include the time taken to enter images into analysis programs or the processing time. User-end time-to-complete is likely similar between the two methods (selecting the images for processing only takes a few minutes), but computer processing time can vary from a few minutes to a few hours but would depend on the complexity of the images, the ease of alignment, and the resolution of the images. Learning curves were slower for this method than for traditional methods; however, after twelve surveys any time differences between the two methods were negligible. These results can be explained by the necessary multitasking required of less experienced snorkelers that need to learn the difficulties of correct identification and record-keeping in the marine environment that are not present in our structure from motion analysis sampling methodology. Traditional methods require the snorkeler to observe, identify and/or count, and to scribe their data, whereas those using the structure from motion method merely focus on correct camera techniques. These results suggest that newer collaborators can become proficient in the analysis of structure from motion surveys at a fast rate and with less difficulty than in traditional methods and that these two methods take similar amounts of time to perform.

Structure from motion analysis from our GoPro technique has benefits among citizen science methodologies for assessing coral reefs in that it can be simplified for non-scientists and allows numerous, accurate assessments for the end-user once the images are processed. Traditional surveys require dive slates, identification guides, and substantial initial training to ensure consistency, whereas GoPro/SFM only requires an action camera, orthogrammic measurements for any reference objects in the imagery, and a simple set of instructions that require minimal to no training. Results can be uploaded to the internet using cloud computing and analysed separately by one or multiple end-users for any desired purpose. It is also possible for citizen scientists to conduct the SFM transformation themselves, but this is a more complex process that requires some online training and was not examined in this research. In addition, many coral reefs or indeed any analogous aquatic benthic habitat can be assessed over a brief period of time on a regional to global scale using this method, thus allowing for broad geographic scale assessments of habitat and the opportunity to build a temporal database of specific sites that could be readily accessible to a range of scientists and projects in the future.

Future studies considering using this methodology should address the following considerations a priori, especially image overlap/resolution and colour correction. Structure from motion requires substantial overlap (>60%) between images for successful alignment (Jebara, Azarbayejani & Pentland, 1999; Torr & Zisserman, 2000). Assuming the rate of movement of a snorkeler is limited, the limiting factor for data sets is, therefore, the depth of the area surveyed. Shallow reefs (<2 m depth) such as the ones in this study require a high image capture rate (2 Hz) to achieve >60% overlap. Due to the increased area covered by the objective, deeper reefs require lower capture rates, though reduced visibility/turbidity can limit the maximum depth-from-substrate. Fewer images per square metre also result in smaller total data throughput, which suggests that transects conducted in deeper waters can cover much larger areas without creating bottlenecks in data analyses (analysing over 1,000 12 megapixel images can use over 32GB of RAM, more than the majority of current-generation PCs, although the recommended hardware to run Agisoft Photoscan includes a quad-core Intel CPU, any compatible motherboard, 16GB of RAM, and a dedicated Nvidia graphics card). The trade-off for deeper reefs would be lowered resolution and greater blue-shift. Surveying from 4 m depth rather than 2 m would most likely quadruple the area covered but lower the resolution by a factor of four, for instance. Given that the maximum resolution achieved at 2 m was 1.2 mm per pixel, however, lowered resolution is likely to become a factor above ∼5 m from the substrate, as resolution would then be lower than 5 mm per pixel. The issue of maximum resolution may be resolved over time when higher megapixel cameras become available. For transects conducted at greater depths by divers or by remote underwater vehicles, users should determine what the end-desired resolution should be (e.g., coral polyps vs. coral volume) and adjust depth-from-substrate accordingly.

One aspect that may separate the two methods is cost. While the initial setup of a GoPro analysis lab is not insignificant (∼$3000 AUD computer, data storage capability, staff to process and analyse transects, program license of $3500 AUD for Agisoft Photoscan though open-access software is available), the benefits of not requiring off-site staff training would quickly recoup the initial costs. Assuming we were tasked with training citizen scientists for traditional snorkelling techniques at the same area we conducted this project (not remote from the Sydney area where the trainers were based), it would take less than six training trips to recoup the costs compared to GoPro citizen science. If surveys are required in remote locations, or over large areas, or over long time-scales, the alternative method presented in this study becomes even more financially viable.

Results of this study suggest that the use of GoPros and structure from motion are reliable for citizen science for assessing sedentary organisms on coral reef habitats. There is a high probability that this GoPro photogrammetry method will also work successfully in other analogous aquatic habitats but this is still to be tested. Our methodology is comparatively low-cost and can be simplified for citizen scientists, yet still allows highly accurate data collection and analyses. Structure from motion is a recent analysis technique limited by current computing power, and the analysis of larger data sets at greater speed will become simpler as more powerful computers and higher resolution action cameras become widely available. The digital nature of the data produced allows for the distribution of the results to a wide research audience for a variety of possible uses.

Supplemental Information

Supplemental Information 1 Total times per transect

Click here for additional data file.

Supplemental Information 2 Results for Coralwatch surveys

Click here for additional data file.

Supplemental Information 3 Benthic diversity data

Click here for additional data file.

Supplemental Information 4 Holothurian counts

Click here for additional data file.

Many thanks to Victoria Richardson for her assistance during experiments, and to the staff at Heron Island Research Station for their logistical support.

Additional Information and Declarations

Competing Interests

Author Contributions

Field Study Permissions

Data Availability

The authors declare there are no competing interests.

Vincent Raoult conceived and designed the experiments, performed the experiments, analyzed the data, wrote the paper, prepared figures and/or tables, reviewed drafts of the paper.

Peter A. David conceived and designed the experiments, performed the experiments, analyzed the data, contributed reagents/materials/analysis tools, wrote the paper, prepared figures and/or tables.

Sally F. Dupont, Ciaran P. Mathewson, Samuel J. O’Neill and Nicholas N. Powell conceived and designed the experiments, performed the experiments, analyzed the data, wrote the paper, prepared figures and/or tables.

Jane E. Williamson conceived and designed the experiments, performed the experiments, analyzed the data, contributed reagents/materials/analysis tools, wrote the paper, prepared figures and/or tables, reviewed drafts of the paper.

The following information was supplied relating to field study approvals (i.e., approving body and any reference numbers):

This research was conducted under Great Barrier Reef marine Park Permit number QC14/004.

The following information was supplied regarding data availability:

The raw data has been supplied as Supplemental Information.

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
