# Peer review of "GoPros™ as an underwater photogrammetry tool for citizen science"

_PeerJ, doi:10.7717/peerj.1960_

## Round 0.1 · original submission · Major Revisions

· Academic Editor

Major Revisions

I now have two reviews of your submission. Both reviewers were positive about the topic and content, but both had reservations about the manuscript in its current form. In particular, Reviewer 2 provides specific suggestions for improvement. As repeated below, please address the reviewers' comments in a point-by-point manner in your rebuttal letter.

Reviewer 1 ·

Basic reporting

In my humble opinion, some literature review of SFM procedures, even if schematic, would enrich the article.

Line 64: I would consider SFM as an image processing technique, not an image analysis.

Line 67: Refrain from using the word mosaic when referring to the 3D model/scene. Mosaicking is a word that is normally related to 2D photomosaicking techniques, not 3D.

Line 69: I’m not sure if photogrammetry can be considered expensive nowadays. But, in any case, I wouldn’t agree that aircrafts are the only vehicles used to gather data (any reference supporting this affirmation?). In fact, I believe that traditionally there have been more land robotics applications using these techniques than aerial ones (which are very new and have gained interest recently because of the growing drone industry).

Line 118: "Snorkelling was done at a slow pace". There is also the issue of rolling shutter related to this topic. The rolling shutter of CMOS cameras such as the GoPro causes the image not to be captured in a single instant, but in scanlines sequentially. This results in a “deformation” of the scene reconstructed using SFM if the camera moves too fast with respect to the surveyed scene. The proposal of performing the image capture process slowly should not only be motivated by the desired overlap between images, but also by the mitigation of the rolling shutter effect. I suggest adding a comment in this direction in this paragraph.

Line 128: I would not describe Photoscan as a “rendering” program. It is a photogrammetry program.

Line 129: “High complexity depth maps were reconstructed from a point map, where the orientation of the mesh was established (approximately 1,000,000 polygons).” The description of the procedure is too vague. Note that this part is NOT part of the structure from motion problem, but from the rest of the photomodelling pipeline.

Line 198: “A supplemental figure 1 is also available for a moving ‘flight’ over the same transect, but using a high polygon 3D model produced from structure from motion.” Where is it available? I did not find it with the supplemental materials.

Line 207: “structure from movement”. This is the only time you refer to the technique with the word “movement”. May be misleading, and is not used in the literature.

Line 264: You can also comment on the dehazing techniques, commonly used in underwater computer vision applications to recover the real colors of the surveyed scene.

Typos:
Line 24: habitats(Belay et al. 2015) --> habitats (Belay et al. 2015) [space missing before parenthesis]
Line 83: with has --> which has
Line 204: surveys were similarly in their assessment --> surveys were similar in their assessment
Line 244: during analysis --> during processing
Line 246: using structure as from --> using SFM as from
Line 313: should be addressed address --> should address
Line 329: remote-underwater-vehicles --> remote underwater vehicles?
Line 344: that this our GoPro photogrammetry method --> that this GoPro photogrammetry method

Experimental design

The experiments are well motivated and sound. Some minor issues that may need further discussion:

As noticed by the authors, the use of commercial software for the SFM has an economic impact on the total cost of the procedure. Being it a research application, the use of freely available software should be considered. Please motivate the use of commercial systems over freely available ones.

Also, it should be stated somewhere that this kind of surveys can be performed with any type of underwater cameras. As you know, due to the popularity of the GoPros, many other action cameras have emerged in the market, and they are as useful as GoPros in performing the proposed task.

Validity of the findings

The differences between the two methods are supported by statistical tests, and show the validity of using the proposed citizen science approach for the tasks described. However, I have some comments on the creation of the SFM models/orthomosaics:

There is no mention to the camera calibration, so I assume it is not calibrated. If the camera is not calibrated, the camera parameters are estimated during SFM. If these parameters are badly estimated, this results on a “bended” surface. This should be especially the case if using GoPros, as they have wide-angle lenses and the images suffer from a lot of distortion whose parameters are difficult to estimate without prior knowledge. If you have entered some kind of information into Photoscan (such as the camera being modelled as a fisheye), you should mention it. As you comment, the surveyed scene is close to planar, so I would like to see a side view of the model, to see if it is really close to the reality or is otherwise bended.

Then, the requirement of SFM techniques to process the data presented is not clear to me. You recover the 3D, but just use the orthomosaic. Wouldn’t a much simpler 2D photomosaicking method do the same job? Please clarify this (maybe the structures in the area under observation are far from planar?).

Line 66: I’m not sure volume measurements can be extracted from the 3D model. Especially in your case, where you are dealing with a bounded surface and there is no limit for the “inner” volume. Did you mean areas? Maybe in this case you are referring to the orthomosaic, but you didn’t mention it up to this point.

Line 249: “studies that count holothurian abundance/diversity should amend the method to use a slow sweeping recording motion using varied GoPro angles to ensure that the areas below corals are effectively covered.” If you map the areas below corals, then you should use the full 3D map to extract conclusions, not the orthomosaic, as the structures below corals will not be visible in this representation.

Reviewer 2 ·

Basic reporting

In this manuscript, the authors evaluate the reliability and performance of a novel data collection protocol, 3D models and associated 2D orthomosaics of 50mx2m transect, to survey several benthic metrics. They compare the results of that protocol to an already established protocol: snorkel survey. The authors also claim that such protocol could be use for ‘citizen science’ efforts, notably because it allows more complete and rich data collection.

3D models of benthic environments built using Structure from Motion are indeed the future of data sampling for most benthic ecologist, especially coral reef ecologist. It is very likely that the upcoming 2016 International Coral Reef Symposium will see the presentation of many scientific results obtained using this tool.

Hence, I agree with the authors when they state that data collected with GoPro photogrammetry will be more easily integrated in future scientific efforts and will allow far more data extraction.
Based on the authors results and my own experience, it makes no doubt to me that such a tool should replace snorkel survey in the future of citizen science, when it is of course both financially and logistically possible.

However, I found the manuscript too weak in its present form to deliver the powerful message the authors are likely aiming for. Indeed, I found parts of the manuscript too vague or inconsistent to allow the reader to really assess the quality and reliability of the presented results (see specific comments). Additionally, some parts of the discussion appeared slightly speculative or not precise enough to me and thus unnecessary.

I would advise for a major revision of that manuscript, and I hope my comments will help the authors in improving the quality of their very timely and needed work.

Experimental design

see below

Validity of the findings

see below

Additional comments

Line 132 “The resulting TIFF files were approximately 60,000 x 20,000 pixels (a resolution of 0.3mm per pixel)”. Hence a 18m*6m rectangle ? I thought the transect were 50m*2m. Please explain.

Line 159 & line 190. The authors twice refer to the use of nested ANOVA, without further explanations. Please provide the complete structure of the ANOVA used here.


Line 172 “..”

Line 192-193 “Time taken for image analysis was not included because it is performed exclusively by the computer software and requires no hands-on input.” Can you provide the reader with at least an order of magnitude? I can see situations in which the time taken to built the orthomosaics will be a non-negligible factor. Please provide more justification for that very important assumption which may greatly influence the results of the protocol comparison.

Line 189-190 “Observer transect times for each transect were then compared using a generalised linear model to estimate rate of improvement for both methods.”
Please develop and provide a more complete description of your modeling effort (in the main text and/or in an appendix).

Line 198-199. “A supplemental figure 1 is also available for a moving ‘flight’ over the same transect, but using a high polygon 3D model produced from structure from motion”
I was not able to find this supplemental material.

Line 206. “While not significant, however, snorkel surveys had a slightly higher degree of variation than structure from movement surveys (Fig. 2).” It is bad practice to describe trends when the test failed to show significant differences. Please remove or reformulate.

Line 215-26. “GoPro photogrammetry allowed large counts of holothurians, and estimates of holothurian abundance were not significantly different for the snorkel transects or the GoPro photogrammetry transects (f = 4.253, df = 1, p = 0.042, Fig 4).”
I would remove the first part of the sentence as it is misleading and thus of poor interest.

Line 242. “limited resolution of the images. Doubling the image resolution (0.15 pixels per mm) may solve this issue, as it would quadruple image size.”
Too speculative and incomplete in my opinion, the quality of the lens and sensitivity of the sensor are of great importance as well. Please develop or reformulate.

Line 244-246. “Despite the limited ability of structure from motion for the identification of coral polyps (due to limited image resolution), estimates of benthic diversity were as reliable using structure as from traditional snorkelling.”
However the apparent systematic bias due to the use of GoPro photogrammetry could be problematic when dealing with a dataset built using both approaches. Please comment.

Line 247-244. “Counts of holothurian abundance were […] was greatly reduced during image analysis as a result”. Too speculative and incomplete in my opinion, how would you then built the orthomosaic ? Will the count of holothuries be done directly with the 3D model ? It needs to be explained in my opinion.

Line 255. The authors compare the use of the CoralWatch health chart with both approaches. They found that results from GoPro photogrammetry were skewed towards the darker color spectrum in comparison to snorkel surveys. However, it appears that the chart and the evaluated coral colony are compared under the same light source during snorkel survey, but different ones when using GoPro photogrammetry. The authors propose to take in account this bias using image color correction. How will this correction be parameterized to allow consistency between different datasets? Wouldn’t it be interesting to randomly drop several charts near corals along the transect to allow for an unbiased used of the CoralWatch health chart approach? It would also improve the scaling of the obtained orthomosaics.

Line 295- Line 301. “Unlike traditional transect methods […]is too complex/expensive for citizen science (McKinnon et al. 2011)”. The two accuracy measurements can not be compared here. The resolution of the orthomosaics does not directly translate to a measure of accuracy. Figueira et al. 2015 provided an actual measurement of precision through a much more refined analysis. I would modify or simply removed this paragraph.

Line 302-312. I found this paragraph of poor interest as it develops a trivial idea. It could only be interesting if an actual accuracy evaluation of the orthomosaics obtained with that material was provided (in a framework similar to Figueira et al., 2015 for example). I would modify or simply removed this paragraph.

Line 313- 331. Too speculative and incomplete in my opinion. How far is this influence by water clarity? What’s the minimum resolution required for the diverse metric presented here? I would modify or simply removed this paragraph.

Line 333-335. Please provide a more detail description of the minimum hardware required for such analysis (CPU, RAM, GPU,HDD,SSD etc.) Also, why would you need Agisoft professional edition over the standard edition ? Additionally, please provide the (much cheaper) price of the educational licence.

---

## Round 0.2 · accepted · Accept

· Academic Editor

Accept

Authors have addressed the reviewers' comments in crafting a revision.